# The Neurosurgical Immigrant Experience in Italy: Analysis of a Northeast Tertiary Center

**DOI:** 10.3390/healthcare13070713

**Published:** 2025-03-24

**Authors:** Andrea Valenti, Elisabetta Marton, Giuseppe Canova, Enrico Giordan

**Affiliations:** 1Department of Neuroscience, Università degli Stud di Padova, 35121 Padova, Italy; andrea.valenti@aulss2.veneto.it (A.V.); elisabetta.marton@aulss2.veneto.it (E.M.); 2Neurosurgical Department, Aulss2 Marca Trevigiana, 31100 Treviso, Italy; giuseppe.canova@aulss2.veneto.it

**Keywords:** immigrant, neurosurgery, healthcare, well-being, spine, brain

## Abstract

Italy’s immigrant population has risen in the last two decades. Integration into society, including access to healthcare, is critical for the well-being of this population. **Objectives**: We compared regular immigrants and Italians to determine whether the groups received different care. **Methods**: Inpatient and outpatient medical records were collected from January 2017 to December 2021. We abstracted the identification code, nationality, sex, age, ICD-9 codes, date of the first and additional visits, and surgical intervention. Pathologies were categorized with ICD-9 codes. Patients were grouped according to geographical origin: European Union (EU), Central and Eastern Europe, Asia, North Africa, Central and South Africa, North America, and Central and South America. **Results**: More patients from Asia and Africa presented to inpatient than outpatient clinics (*p*-value: 0.001). The median age was lower for patients from Asia and Eastern Europe than from the EU. More patients presented with acute spine pain (26.4% versus 19.6%, *p*-value: 0.001) as inpatients, while patients presented as outpatients more for degenerative spine issues (77.1% versus 69.0%, *p*-value: <0.001) but less for brain neoplasms (*p*-value: 0.009). Additional visit rates were higher for immigrants than for Italians (IRR 1.32 visits/year, 95% CI 0.99–1.77 visits/year, *p*-value: 0.06), especially for patients with spinal issues (spinal versus cranial: 1.27 visits/year, 95% CI 1.14–1.43 visits/year, *p*-value: <0.001) and younger patients (<65 years old: 1.52 visits/year, 95% CI 1.39–1.71 visits per year, *p*-value: <0.001). There was no difference in the incidence of new visits when stratified by sex. **Conclusions**: Access to emergency care and additional visits were more prevalent in the recent immigrant population, especially from Asia, reflecting unconsolidated health habits. Immigrants from Central and Eastern Europe or North Africa seemed fully integrated. A healthcare policy tailored to the needs of immigrants—taking into account their cultural and social backgrounds and ensuring effective communication—can be highly beneficial. Specifically, it is essential to reintegrate general practitioners and guide individuals toward the most appropriate services.

## 1. Introduction

Italy’s immigrant population is rapidly increasing and has been rising over the last two decades. In 2002, 1.3 million of Italy’s inhabitants were immigrants; by 2014, this number had increased to 4.9 million and was 5.2 million by 2019 [1,2]. Immigrants account for nearly 10% of Italy’s population.

Integration into society, including access to healthcare, is critical for the well-being of these individuals. Moreover, there is a trend in immigrants experiencing poorer health the longer they stay in a country, thus increasing their need for access to healthcare [3]. A recent study found that, in Italy, regular immigrants have 29% increased odds of having unmet healthcare needs. In contrast, irregular immigrants (i.e., those who do not meet the conditions for entry or stay) have 59% increased odds of having unmet healthcare needs [4]. Language and the structure of healthcare systems can significantly impact the ability to meet the needs of immigrants. These needs are complex and can vary depending on who is defining them and the potential for recall bias. Self-reported levels of unmet healthcare needs are often influenced by the most significant or distressing memories individuals have. This situation is commonly observed in other countries due to cultural, socioeconomic, and communication barriers that arise from different habits and practices [5,6,7].

The Italian Constitution (Article 32) guarantees that health is a right for all: “The Republic safeguards health as a fundamental right of the individual and as a collective interest and guarantees free medical care to the indigent”. However, there are barriers to healthcare access among Italy’s migrant population. Understanding these barriers to access and their impact on health is the first step in understanding how to improve access to healthcare for this population.

This study will review patients visiting the Department of Neurosurgery in a tertiary level hospital in north-east Italy (Treviso, Italy) from 1 January 2017 until 31 December 2021. We analyzed two groups: regular immigrants and Italian patients. The primary objective of this study is to determine whether immigrant and native Italian populations receive different patient care for brain and spinal conditions. The data collected in this study will allow us to better understand how we serve immigrant and Italian patients.

Discussing healthcare by comparing immigrants and native people in neurosurgery appears quite interesting because it is a very specialized sector. Furthermore, every center can provide a wide range of patients with the same criteria in diagnosis, treatment (conservative or surgical), and follow-up. Moreover, neurosurgery deals with the brain and the spine, so with a kind of disease that is perceived as urgent and unpreventable. Immigrants are not accustomed to discuss health problems with their general practitioner. So, by failing to provide adequate prevention, neurosurgical evaluations multiply unnecessarily, either as an outpatient or inpatient, through the emergency department.

## 2. Materials and Methods

### 2.1. Data Source

Data were retrospectively abstracted from our institution’s centralized healthcare inpatient and outpatient management system (PMS). The PMS collects and maintains a repository of all ongoing medical data pertinent to the care of Aulss2 Marca Trevigiana patients.

In the present analysis, we utilized the PMS resources to identify all patients who presented with neurosurgical issues over the last five years (1 January 2017–31 December 2021). The primary search found 43,194 patients that required a neurosurgical visit. Outpatients numbered 26,402, while inpatients numbered 16,792.

Inpatients were those who presented to the Emergency Care Unit, possibly requiring immediate neurosurgical care, and those who accessed emergency care, were later hospitalized in other units, and subsequently required neurosurgical care. Outpatients were those patients who visited the clinic electively for diagnosis, chronic disease evaluation, or treatment planning.

Regarding the distribution of immigrants in the Treviso area, data from the Istituto Nazionale di Statistica (ISTAT) showed that immigrants accounted for 10.5% of Treviso residents on 1 January 2021. In this area, the majority of immigrants come from Central and Eastern Europe (28.4%), followed by Asia (18.8%), South Africa (11.1%), North Africa (10.4%), and Central and South America (4%). The remaining immigrant patients were from European Union (EU) countries (27.0%) and North America (0.3%) [1,2].

The largest foreign community comes from Romania, with 22.4% of all foreigners in the area, followed by the People’s Republic of China (10.3%) and Morocco (9.1%). The immigrant population is almost evenly distributed by sex (50.7% of male immigrants, male-to-female ratio 1.03:1). Most immigrants were under 65 years of age (91.6%), while the proportion of people <65 years old in the resident population was 70.7% [1].

The data from ISTAT are summarized in Figure 1 and Table 1.

### 2.2. Study Criteria and Data Collection

The IRB approved the study at our institution (Aulss2 Marca Trevigiana), and only patients who gave signed consent for using their information were included. Aulss2 Marca Trevigiana is the neurosurgery referral center for the Treviso district, covering more than 1 million patients.

Patients of any age and sex accessing our clinics were abstracted from 1 January 2017 to 31 December 2021. An experienced data abstractor provided the complete inpatient and outpatient medical records of all patients who sought neurosurgical care in the past five years. An investigator reviewed all patients initially screened by the abstractor (E.G.). The data abstracted were the following: national identification code, nationality, sex, age, diagnosis by ICD-9 codes, date of the first visit, date of subsequent visits, and data on surgical interventions.

Those who were born in another country and then moved to Italy and gained a national identifier code were considered immigrants. This gave them access to the Italian national healthcare system. Second-generation immigrants (i.e., those born and raised in Italy with at least one foreign-born parent) were excluded from the immigrant cohort. This was designed to capture only foreign-born or first-generation immigrants and, consequently, an actual picture of the change in the perception of healthcare systems.

ICD-9 codes were utilized to categorize patients based on their pathologies. Initially, patients were divided based on the issues they presented: cranial, spinal, peripherical, or neurological.

Cranial pathologies included brain neoplasms (metastasis, pituitary adenomas, gliomas, meningiomas, etc.), congenital diseases (craniosynostosis, Chiari malformations, etc.), vascular malformations (aneurism, arteriovenous malformation e/o fistulas, cavernomas, etc.), acute or treated post-traumatic brain conditions (chronic subdural hematomas/acute hematoma, intraparenchymal hematoma, etc.), hydrocephalus or cerebrospinal fluid (CSF)-related issues (e.g., shunt malfunction), and infection-related conditions (e.g., brain abscess).

Spinal pathologies included degenerative spine conditions (herniations, disc degeneration, deformity, spondylolisthesis, etc.), vascular malformations (dural or epidural fistulas), neoplasms (meningiomas, schwannomas, gliomas), and acute or chronic fractures (even post-treatment fracture control).

Peripheral pathologies included peripheral nerve schwannoma and median or ulnar compression syndromes.

Patients with neurological conditions attended our clinics for headaches, migraines, or stabilized peripheral nerve syndromes, which essentially do not require surgical interventions.

Additionally, we abstracted more detailed diagnosis types based on ICD-9 codes, such as post-traumatic brain and spine conditions, disc herniation, spine degenerative diseases (i.e., stenosis, spondylolisthesis, black disc, juxtafacet cysts), neoplasms (any benign or malignant brain neoplasms), vascular malformations, and hydrocephalus-related diseases (i.e., idiopathic normal pressure hydrocephalus).

The diagnosis was made using the same tools applied to the Italian patients. First, we investigated the patient’s medical history, which guided the subsequent physical examination. Then, we clarified diagnostic suspicions with instrumental analysis, such as MRI, CT scans, X-rays, neurophysiology tests, blood tests, and other specialist assessments.

The type and distribution of diagnosis changed based on inpatient versus outpatient presentations. Inpatients typically suffer from severe medical conditions that progress quickly, necessitating urgent hospitalization and prompt diagnosis, which may include surgical intervention. In contrast, outpatients usually have conditions that progress more slowly, allowing for treatment to be scheduled later based on their position on the waiting list and the nature of their disorders.

Immigrants were categorized by geographical origin: the European Union (EU) (i.e., immigrants from other EU countries), Central and Eastern Europe (i.e., Albania, Bulgaria, Croatia, the Czech Republic, Hungary, Poland, Romania, the Slovak Republic, Slovenia, and the three Baltic States: Estonia, Latvia, and Lithuania; Eurasian states were also included, such as Armenia, Azerbaijan, Belarus, Georgia, Kazakhstan, Kyrgyzstan, Moldova, Russian Federation, Tajikistan, Turkmenistan, Ukraine, and Uzbekistan), North Africa (i.e., Libya, Morocco, Egypt, Algeria, Sudan, and Tunisia), Central and South Africa (i.e., Central and Sub-Saharan African countries, such as South Africa, Eritrea, Somalia, Ethiopia, Mali, Niger, Congo, Tanzania, and Senegal), North America (i.e., Canada, Alaska, and the United States), and South and Central America (i.e., Mexico and all South American countries). These macro groups, among the included countries, share similarities in cultural, socioeconomic, and linguistic backgrounds, enabling synthetic yet effective data collection.

To capture the migratory flux from Central and Eastern Europe with a high degree of precision, we considered only EU states that have been members for at least ten years prior to the inception of the study (2007). All the other countries were considered non-EU countries and grouped in the Italian versus immigrant analysis.

Cultural habits influence the perceived needs in health and subsequent decisions [5]. Immigrant women, particularly those from Asian and South Asian backgrounds, as well as practicing Muslims, seek female physicians for their healthcare needs [8,9,10]. They prefer physicians from the same ethnic background, thinking they better understand cultural and religious norms [10]. Immigrant women from South Asia, Eastern Europe, Africa, and Central and South America often feel shy and uncomfortable during physical exams due to their adherence to traditional values regarding modesty and the exposure of body parts [11,12]. Sometimes immigrant women (from Asia, South Asia, and Muslim countries) do not feel comfortable sharing their health problems with their family members or friends due to fears of being ostracized. In many countries, having a severe disease or mental illness is interpreted as a matter of shame and disgrace to the family [13,14].

### 2.3. Statistical Analysis

For continuous variables, descriptive statistics were reported as the median and interquartile range (IQR), and for categorical variables, the proportion as a percentage. Categorical variables were analyzed using the Chi-square test or Fisher’s exact test. When necessary, annual incidence rates were calculated for Italian and immigrant populations stratified by age, sex, and disease (spinal versus cranial) strata. Confidence intervals (CIs) were estimated via the Poisson distribution. Zero-inflated Poisson regression was used to assess the relationship between new visit incidence rates and sex, age, and presenting issues for either inpatients or outpatients, adjusting the analysis for age (underage and adult) and sex. This process was conducted for each geographical macroarea as well as for individual countries within those areas.

All statistical tests were two-sided, the type I error rate was 0.05, and analyses were conducted using Stata (version 13.0, StataCorp: College Station, TX, USA).

## 3. Results

During the study period (2017–2021), 43,194 patients visited our inpatient and outpatient clinics. Outpatients numbered 26,402, while inpatients numbered 16,792. The final cohort excluded patients from whom we were unable to collect complete personal and clinical data, irregular immigrants, or patients who did not meet the inclusion criteria (2756 or 10.4% for outpatients and 3124 or 18.6% for inpatients). The final sample consisted of 37,313 patients.

In the inpatient group, 89.9% were Italian and 10.1% immigrants. In the outpatient group, 86.8% were Italian, and 13.2% were immigrants.

Most Italian inpatients were ≥18 years old (95.7%), with only 4.3% of them under the legal age. In the immigrant population, only 2.7% of inpatients were under 18. More than one-third of Italian inpatients were over 65 (40.7%). In the outpatient group, most Italian patients were ≥18 years old (96.0%), with only 4% underage. In the immigrant population, only 1.5% of outpatients were under 18. Nearly half of the Italian outpatients were under 65 (47.9%), while only 13.6% of the immigrants were under 65.

There was a significant difference in the percentage of patients <65 and <18 years old between the patients who were Italian and those who were immigrants (40.7% versus 81.7%, *p*-value: 0.001 and 47.9% versus 86.3%, *p*-value: <0.001; 4.3% versus 2.7%, *p*-value: 0.038 and 4.0% versus 1.5%, *p*-value: <0.001, respectively) in both the inpatient and outpatient groups, with a significantly higher percentage of younger patients in the immigrant group presenting for inpatient and outpatient visits. Furthermore, a greater proportion of <18-year-old immigrants presented as inpatients than as outpatients (2.7% versus 1.5%, *p*-value: 0.025), but this was not the case in the Italian populations.

Overall, the younger population group in the non-EU community was of Asian origin, where only 4.9% of patients were younger than 65 years old. In contrast, the population with the highest proportion of >65 patients presenting at our outpatient clinic was South American (20.4%), followed by North African (10.2%).

In the inpatient immigrant group, the majority of patients were originally from Central and Eastern Europe (40.5%), followed by patients from EU countries (19.2%), Asia (12%), Central and South America, (9.3%), North Africa (8.3%), South Africa (4.3%), and North America (1.4%) (Figure 1B).

In the outpatient immigrant group, the majority of patients were originally from Central and Eastern Europe (52.0%), followed by patients from EU countries (15.1%), North Africa (10.7%), Asia (7.4%), Central and South America (8.3%), South Africa (5.3%), and North America (1.2%) (Figure 1C). A significantly higher percentage of patients from Asia and South Africa presented to inpatient clinics than outpatient clinics (12% versus 7.4% and 9.4 versus 5.4%, *p*-value: 0.001, respectively).

The median age at presentation for Italian inpatients was 63 years (IQR range 50–75 years), and there was a slight male predominance (52.3%; male-to-female ratio 1.1:1). Overall, the median age at presentation for immigrant patients was 51 years (IQR range 42–59 years), and there was a slight male predominance too (51.7%; male-to-female ratio 1.07:1). The male-to-female ratio was similar between Italians and immigrants from different geographical areas, with the exception of North African and South African patients, who had a pronounced male predominance (71.6% and 65.5%, respectively).

The median age at presentation for Italian outpatients was 63 years (IQR range 50–75 years), and there was a slight male predominance (52.3%; male-to-female ratio 1.1:1). The median age at presentation for immigrant patients was 51 years (IQR range 42–59 years), and there was a slight male predominance (51.7%; male-to-female ratio 1.07:1). After stratifying the immigrant group by country of origin, the median age was found to be lower for Asian and Central and Eastern European patients (47 years, IQR range 39–56 years, and 49 years, IQR 41–57 years, respectively) compared to EU ones (58 years, IQR 52–64 years).

The demographic characteristics are summarized in Table 1.

Most of the Italian inpatients were referred to our inpatient clinic for cranial-related problems (55.0%), followed by spinal issues (44.1%), neurological conditions (0.7%), and peripheral nerve-related issues (0.3%). Spine- or brain-related control visits accounted for 11.5% of the visits by the Italian cohort. Most of the immigrant inpatients were referred to us for cranial-related pathologies (49.6%), closely followed by spine-related problems (49.4%), neurological conditions (0.6%), and peripheral nerve-related issues (0.4%).

There was a significant difference in the distribution of cranial- and spinal-related visits, which were almost evenly distributed among immigrant inpatients, but there was a higher number of visits for cranial issues in the Italian population (*p*-value: 0.023).

Most of the Italian outpatients were referred to our outpatient clinic for spine-related problems (76.6%), followed by cranial issues (21.3%), peripheral nerve-related issues (0.9%), and neurological conditions (0.3%). Most of the immigrant outpatients were referred to us for a spine-related problem (82.1%), followed by cranial issues (16.3%), peripheral nerve-related issues (1.3%), and neurological conditions (0.4%). Italian outpatients were more likely to present with cranial issues (*p*-value: <0.001), whereas immigrant patients were more likely to present with spinal pathologies (*p*-value: <0.001).

After stratifying the reasons for inpatient visits, we found that the majority of Italian inpatients presented with traumatic conditions (57.3%, 9.8% with spinal fractures), followed by 19.6% with acute pain due to degenerative spine conditions, 12.4% with brain neoplasms, 6.2% with vascular accidents, and 4.5% with brain and spine infections or CSF circulation problems. The majority of immigrant inpatients presented with traumatic conditions (55.5% overall, 9.1% with spinal fractures), followed by 23.7% with acute pain due to degenerative spine conditions, 10.5% with brain neoplasms, 5.4% with vascular accidents, and 4.8% with brain and spine infections or CSF circulation problems.

The differences in inpatient distribution are even more evident when only non-EU immigrants are analyzed, with a significantly higher percentage of patients presenting with acute pain due to degenerative spine conditions (26.4% versus 19.6%, *p*-value: 0.001).

The majority of Italian outpatients presented with degenerative spine conditions (69.0%), followed by 15.8% for brain and spine control visits, 7.7% with brain neoplasms, 4.0% with brain and spine infections or CSF circulation problems, and 3.5% with vascular malformation diagnoses. The majority of immigrant outpatients presented with degenerative spine conditions (76.5%), followed by 15.5% for brain and spine control visits, 7.7% with brain neoplasms, 2.8% with brain and spine infections or CSF circulation problems, and 2.3% with vascular malformation diagnoses.

The differences in outpatient distribution are even more evident when only non-EU immigrants are analyzed, with a significantly higher percentage of patients presenting with degenerative spine issues (77.1% versus 69.0%, *p*-value: <0.001) and a lower percentage of patients presenting with brain neoplasms (*p*-value: 0.009).

Additional visit incidence rates were higher for immigrant patients than for Italian patients (IRR 1.32 visits/year 95% CI 0.99–1.77 visits/year, *p*-value: 0.06). Such a rate is even more evident in the immigrant population after stratifying the visit rate by the issue presented (spinal versus cranial: 1.27 visits/year, 95% CI 1.14–1.43 visits/year, *p*-value: <0.001) and age (<65 years old: 1.52 visits/year, 95% CI 1.39–1.71 visits per year, *p*-value: <0.001). There was no difference in the incidence of new visits when stratified by sex.

The median intervals between visits for Italian outpatients with cranial and spinal issues were 18.6 months and 17.7 months, respectively. In contrast, median intervals between visits for immigrant outpatients were 15.8 months and 14.8 months.

Less than one-quarter (23%) of Italian inpatients underwent a surgical procedure compared to 18.9% of immigrant inpatients (*p*-value: 0.049)—specifically, 11.3% of non-EU patients, 11.4% of Asian patients, 10.9% of Central and Eastern European patients, 10.6% of South American patients, 13.7% of South African patients, and 11.4% of North African patients. Overall, 17.5% of Italian patients underwent surgery after presenting with cranial diseases, while 7.8% had spinal issues.

In the end, 6.9% of Italian outpatients underwent a surgical procedure compared to 4.9% of immigrant inpatients (*p*-value: 0.035)—specifically, 11.3% of non-EU patients, 11.4% of Asian patients, 10.9% of Central and Eastern patients, 10.6% of South American patients, 13.7% of South African patients, and 11.4% of North African patients. Overall, 17.5% of Italian patients underwent surgery after presenting with cranial diseases, while 7.8% had spinal issues.

An analysis of inpatient versus outpatient distribution is presented in Table 2.

## 4. Discussion

Increased research focus has been placed on differences in healthcare use between immigrant and non-immigrant groups. Nevertheless, to our knowledge, neurosurgical care between native and immigrant populations has never been the subject of research. Over the years, a large body of literature has consistently shown that immigrants risk unequal access to appropriate healthcare [15,16]. Even in countries where access to healthcare is guaranteed to all people, as in Italy, immigrants could experience barriers raised by individual, sociocultural, economic, administrative, and political issues [16]. Our study revealed that the distribution of immigrants accessing public healthcare for neurosurgical issues is comparable to that in the population of Treviso. As expected, immigrant patients are significantly younger than Italian ones, with Asians and South Americans being the youngest to seek inpatient and outpatient care. In contrast, Central and Eastern European immigrants, followed by North African immigrants, are the oldest, possibly reflecting the older migratory flux into the Treviso area.

When stratified by geographical area, we noticed that despite being the second most represented immigrant cohort in the population, Asian immigrants were less prevalent in outpatient services. In contrast, they are the second most common group of immigrants in inpatient visits. Inpatient and outpatient visits are dominated by immigrants from Central and Eastern European countries. Immigrant patients seek emergency care for acute spinal diseases more frequently than their Italian counterparts. The same trend is highlighted for outpatient visits, where spine issues are most prevalent.

Regarding the need for additional visits, we highlight that the immigrant population has higher rates of additional visits for the same problem than the Italian population, particularly among younger patients and those with spinal diseases.

In the end, the proportion of patients who underwent surgical intervention does not differ between immigrant and Italian patients, anyway there are slight but significant differences when considering non-EU ones.

Immigrants often face a higher prevalence of manual labor or low-skilled jobs, temporary or informal employment, and lower wages. They tend to experience greater physical demands, poorer environmental working conditions, and increased exposure to occupational risks, particularly ergonomic and psychosocial hazards. Additionally, they generally have worse overall mental health as well as challenging employment and working conditions, a situation that the economic crisis has exacerbated [17,18]. All of these characteristics are quite common in most Western countries [5,6].

Overall, from our results, we can speculate that there are no apparent differences in neurosurgical conduct between native and immigrant patients and that the differences in access to care distribution are speculatively inherent in the different geographical origins and timing of migratory flux. We highlighted how Central and Eastern European and North African immigrants had reached a very similar distribution compared to that of Italian patients seeking neurosurgical care. This is likely due to the time those populations have spent in Italy in the past decades. This may allow them to adapt to the different healthcare systems. In the case of Asian immigrants, who are relatively new to the area, they possibly have not had enough time to fully integrate into the healthcare services, develop a relationship with a general practitioner, or discard some of the health-related beliefs of their culture. This likely results in higher emergency care use and lower outpatient service utilization [19]. In fact, the use of emergency rooms or services is considered a form of primary care, and it has been associated with a lack of health insurance, a lack of access to primary care, low socioeconomic status, the patient’s perception of the severity of health issues, and convenience [20].

Additionally, we highlight the possibility that immigrants are marginally less likely to accept surgical interventions than Italians. This may be reflective of linguistic barriers and, thus, insufficient communication or because of cultural beliefs that instill fear of invasive procedures [20,21]. In the end, inadequate communication or an inability to understand physician instructions can account for the immigrant population’s higher incidence of additional visits for the same issues over the years [19]. This is particularly true for visits for young patients with spinal issues; this could be related to the fact that immigrants are often employed in jobs requiring a large amount of physical labor, thus they seek solutions for spine-related pain that can affect their working ability. Additionally, such subpopulations may not be able to access ancillary services such as pain or phychinekinesis therapy [19].

It has been proven that the odds of experiencing unmet needs are higher for regular immigrants than for Italian citizens. These gaps are even more striking in the sub-sample of those suffering from chronic health conditions [4]. Although the Italian national health system guarantees access to health and social services for the immigrant population, there are inevitable social and territorial inequalities in guaranteeing basic or advanced levels of care [15,22,23].

It is well-known that healthcare response can be complex due to cultural, religious, and linguistic differences in individuals exposed to specific risk factors before, during, and after migration [15,24]. The literature identifies various reasons to explain the differences between native and foreign populations. The most significant barriers include cultural factors, communication issues, socioeconomic status, the structure of the healthcare system, and immigrants’ knowledge of the healthcare system [5,22]. Among these, there is also perceived discrimination and racism [25], difficulty accessing the system, longer wait times, and a lack of direct access to specialists, laboratories, or other diagnostic reporting [17]. Cultural factors should not be underestimated, such as the inability to express personal beliefs like previous immigrants did or the need to adapt to certain aspects of the Western lifestyle, including changes in diet and the work–leisure balance [26]. Hence, immigrants tend to use healthcare services less frequently than native residents, terminate treatment early, and receive lower-quality healthcare [6].

There is also an inherited negative attitude toward interacting with the host country’s health system [15,27]. Research indicates that immigrants often have better health than native residents when assessed through self-reports and health indicators upon their arrival in the host country. However, over time, this phenomenon known as the “healthy migrant effect” diminishes, and their health tends to decline [28,29]. Some immigrants are also susceptible to poor health due to exposure to numerous health hazards before, during, and after immigration [6].

Despite contributing to a productive and skilled society such as the Italian one, immigrants often lack knowledge of national laws and administrative regulations, eventually suffering from precarious living conditions and social exclusion [15,30,31]. This complicates the ability of immigrants to establish relationships with physicians and, as highlighted by our results, receive better emergency care than in outpatient clinics or from general practitioners. In population-based cohort studies, it was found that recent immigrants were 1.2-fold more likely to use emergency care services at the end of life and 1.1-fold more likely to die in acute care settings than long-standing residents [17].

The literature also demonstrates the importance of sex for understanding immigrant inequalities. It was found that women are often more likely to experience barriers to accessing care than men. The same gender gap was noted in public and private healthcare systems [32,33]. Such difficulties were found especially for pregnant women who found barriers to accessing pre-natal and maternal healthcare due to a lack of awareness of their rights and the provision of healthcare in the host country [33]. Such differences in sex distribution were not found in the neurosurgical population for cranial or spinal issues.

Furthermore, a large Italian cohort study showed higher avoidable hospitalization rates among adults from countries with strong migratory pressure [34]. In line with our study, these findings seem to confirm the lower utilization of outpatient and specialist care among recent immigrants [30]. Additionally, the hospital is often the only or the main point of access to healthcare services in many immigrants’ countries of origin [15,32]. Furthermore, studies in countries with insurance-based healthcare systems confirm that immigrants have greater difficulty accessing care, leading to greater use of acute care [34]. It can be hypothesized that immigrants are not always aware of other healthcare resources or facilities, probably because navigating the complex bureaucracy, compounded by language barriers, makes the service inaccessible [15].

There is often a lack of knowledge about immigrants’ health needs, which frequently arises from communication barriers. Access to interpreter services along with a deeper understanding of health practices in their home countries—particularly in developing nations—can significantly enhance the effectiveness of medical interventions. This improvement is significant during residency programs [31].

Health policies must address the diverse needs of immigrants. Collaborating with non-governmental organizations (NGOs) can help achieve this goal as long as these organizations employ personnel trained to understand immigrants’ cultural, socioeconomic, and educational backgrounds. Integrating immigrants into society is crucial for guiding them toward appropriate health services.

Schools can play a vital role by offering mandatory health courses that educate younger generations of immigrants. This knowledge can then be shared within families, benefiting older relatives and those recently arriving in Italy. Additionally, family doctors are essential in directing patients to the appropriate exams and specialist visits, which can help reduce reliance on emergency care [24].

Emergency departments are designed to handle acute symptoms but often do not assess the overall clinical picture. This results in patients frequently returning with the same symptoms without a thorough outpatient diagnostic process. Such repetition can lead to increased healthcare costs, borne by taxpayers in Italy, and longer waiting times in emergency departments [32,34].

Moreover, this reliance on emergency care often exacerbates health issues among immigrants, leading to silent disease progression, such as tumors, chronic pain in spinal disorders, and a greater need for invasive surgeries in cases of severe fractures. Complex health conditions can result in incomplete recovery or increased surgical complications due to decreased mobility, affecting patients’ ability to return to their daily activities.

In the end, in this study, due to the applied methodology based on national identifier codes, we could not abstract data on irregular immigrants, who are a small percentage of immigrants but still require health attention. Such a subgroup came to our attention mostly in acute settings for severe invalidating brain and spine problems, which often require surgical attention, opening post-operative care settings, and reimbursement difficulties for the hospital. Studies that try to include such subgroups of patients are required to understand the immigrant phenomenon better.

### Limitations

A significant limitation of this study is that we could not conduct a more detailed sub-analysis of sub-pathologies based on ICD-9 codes, which could have explored interactions with factors such as country of origin, age, and sex. Additionally, the data abstraction did not include socioeconomic status or patients’ income. The inability to perform further analysis considering these modifiers and confounders may have hindered a deeper understanding of the overall analysis. Additionally, a significant limitation is that we could not include irregular immigrants in our analysis. In Italy, once immigrants enter the country to live, they acquire an identification code that uniquely identifies them within the national healthcare system and for other services, including taxation. We utilized this identification code to trace the original nationality of immigrants, which means that these are all regularly registered immigrants. Once they are registered with a national identification number, it is irrelevant if they are irregular or regular migrants, and they have access to the same healthcare. Therefore, no irregular immigrants were included, or rather, only regularized ones were fully extracted.

Also, we could not track the duration of stay for these immigrants in our country. This is because our healthcare system does not keep records of this information during inpatient or outpatient visits. Our healthcare system is based on the principles of the Italian Constitution and is funded by citizens’ taxes. Therefore, there is no requirement for insurance, and we have no interest in knowing a patient’s disposable income, except for those living abroad, who are not the focus of this study.

There is still a significant percentage of the population whose data we cannot abstract properly. Another limitation is that we could not abstract how many years the immigrants had spent in Italy. This could have helped in understanding the impact of time on adjusting to a new health system. Furthermore, we did not have information about immigrants’ satisfaction and comprehension (i.e., the language spoken and with what proficiency) after a visit, either in inpatient or, more specifically, outpatient services. Such information could have helped in understanding the cultural and language barriers that could limit access to adequate neurosurgical care. In the end, the geographical area of the study limits its generalization to another part of Italy due to slight but unavoidable cultural and healthcare differences between Italian regions. It must be considered that the Treviso district is quite a rich area, with the highest GDP per capita in Veneto, and Veneto has the third highest GDP in Italy. This makes the area one of the wealthiest in Italy and Europe, with full access to healthcare. This may lead to differences in patient conduct considering the availability of resources. However, this is the first study to analyze such a phenomenon in a neurosurgical setting. More prospective studies are needed, along with more specific questionnaires, to understand in more detail what the real needs of immigrants are in such a complex setting as neurosurgical care.

## 5. Conclusions

There were no clearly evident differences in conduct between inpatient and outpatient care in immigrant patients compared to Italians. Only slight differences were found in accessing emergency care and additional visits, which are more prevalent in the recent immigrant population, especially among those from Asia, and reflect unconsolidated health habits in those populations. This highlights the importance of the basic health education provided by schools and general practitioners. An effective health policy should encourage investments to enhance the socioeconomic conditions of immigrants in collaboration with non-governmental organizations. Older immigrants, especially from Central and Eastern Europe, were fully integrated into the healthcare system. Attention should be given to young immigrants with spine issues, for whom appropriate communication and care may avoid recurring presentations for the same problem.

## Figures and Tables

**Figure 1 healthcare-13-00713-f001:**
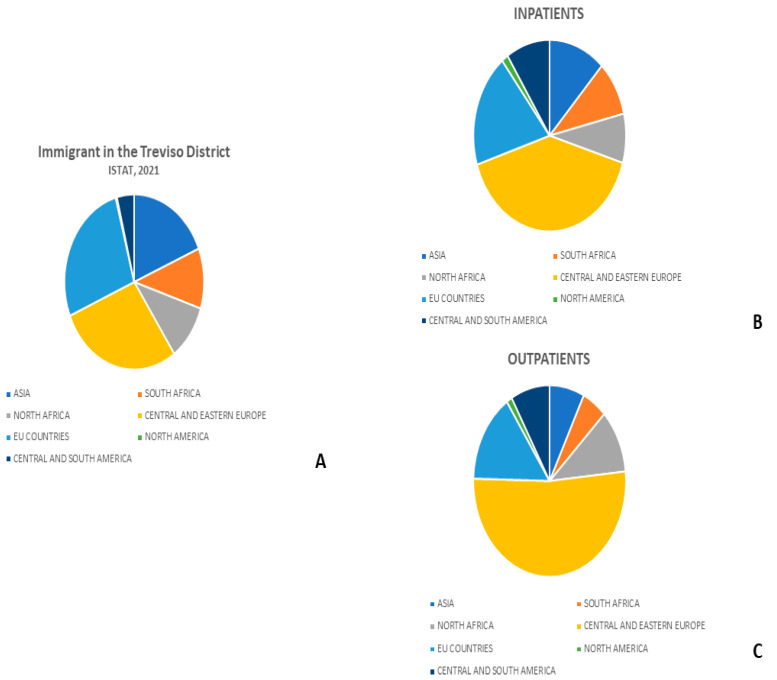
Distribution of immigrants by country of origin in the Treviso district and in our neurosurgical inpatient and outpatient clinics. Legend: (**A)** Immigrants in the Treviso district; (**B**) Inpatients; (**C**) Outpatients.

**Table 1 healthcare-13-00713-t001:** Demographic characteristics and distribution of regular immigrants in the Treviso district and neurosurgical inpatient and outpatient clinics. Legend: TOT: total, N°: number, IQR: interquartile range, *: data from ISTAT, 1 January 2021.

	Treviso District *	Inpatients			Outpatients		
	Immigrants	Italians	Immigrants	*p*-Value	Italians	Immigrants	*p*-Value
N° **TOT**	92.110	16.792			26.402		
**% TOT**	10.5%	89.9%	10.1%		86.8%	13.2%	
**Sex**	49.3%	54.3%	52.2%	0.379	49,10%	49.8%	0.466
(female)							
**Age**		70 (54–81)	51 (39–61)	<0.001	63 (50–75)	51 (42–59)	<0.001
(years, median, IQR)							
**Age < 65**	91.6%	40.7%	81.7%	<0.001	47.9%	86.3%	<0.001
(years)							
**Age < 18**	23.4%	4.3%	2.7%	0.038	40.0%	1.5%	<0.001
(years)							
**Country of Origin**							
European Union	27.0%		19.2%			15.1%	
Asia	18.8%		12.0%			7.4%	
North Africa	10.4%		8.3%			10.7%	
South Africa	11.1%		9.3%			5.4%	
Central & South America	4.0%		9.3%			8.3%	
Central & Easter Europe	28.4%		40.5%			52.0%	
North America	0.3%		1.5%			1.2%	

**Table 2 healthcare-13-00713-t002:** Distribution of inpatient and outpatient visits by pathology and sub-pathology. Legend: **EU**: European union.

	**Inpatients**					**Outpatients**		
	**Italians**	**Immigrants**	***p*-Value**			**Italians**	**Immigrants**	***p*-Value**
**Pathology**								
Cranial	55.0%	49.6%	0.023			21.3%	16.3%	<0.001
Spinal	44.1%	49.4%	0.025			76.6%	82.1%	<0.001
Peripheral	0.3%	0.4%	0.226			0.9%	1.3%	0.074
Neurological	0.7%	0.6%	0.407			0.3%	0.4%	0.438
**Sub-Pathology**								
Extra EU Immigrants								
Trauma	57.3%	54.7%	0.548	Spine or Brain Conditions Control Visits		15.8%	12.3%	0.176
Spinal Degenerative Conditions	19.6%	26.4%	0.001	Spinal Degenerative Conditions		69.0%	77.1%	<0.001
Brain Neoplasms	12.4%	9.5%	0.090	Brain Neoplasms		7.7%	5.3%	0.003
Vascular Accidents	6.2%	5.2%	0.425	Vascular Accidents		3.5%	2.3%	0.009
CSF or Infectious Issues	4.5%	4.2%	0.921	CSF or Infectious Issues		4.0%	3.0%	0.721

## Data Availability

Data are contained within the article.

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
