# Peer review of "The Neurosurgical Immigrant Experience in Italy: Analysis of a Northeast Tertiary Center"

_healthcare, 2025, doi:10.3390/healthcare13070713_

Round 1

Reviewer 1 Report

Comments and Suggestions for Authors

I reviewed the manuscript submitted by Valenti et al titled - The neurosurgical immigrant experience in Italy: analysis of a northeast tertiary center. This manuscript represents a neurosurgical care analysis of the access of immigrants to a tertiary center in northeastern Italy and the differences in the utilization of health care by immigrant compared to native Italian populations. This paper shows significant demographic & diagnostic disparities present. This is pointing out that the immigrant patients are younger, with a higher incidence of spinal pathologies & emergencies. This is a significant observation to be reported. It also highlights that immigrants, especially those coming from Asia and Central/Eastern Europe; have higher follow-up visit rates, which may reflect potential communication problems or unfamiliarity with the health system. The study underlines how complex the needs of immigrant healthcare are, pointing at cultural, linguistic, and socioeconomic factors influencing access and results. It is well-pegged with data and points to some critical challenges on gaps in integration for recent immigrants, including higher rates of utilization of emergency care as a substitute for primary care.
I must say that this study is quite broad, but it has its limitation in not being able to include irregular immigrants or stratify data based on duration of stay, which could further contextualize the trends of integration.
The manuscript would be further enhanced with minor clarifications on statistical methodologies and further exploration of cultural barriers.
Authors are advised to take care of these comments to make the article better.

Author Response

Thanks for reviewing our manuscript: The neurosurgical immigrant experience in Italy: analysis of a northeast tertiary center.. Manuscript number: Healthcare-3438873

We have made each of the changes suggested by the reviewers in red and bold, and a copy of the revised manuscript is enclosed for your review.

Please see below for a point-by-point response to the reviewer’s comments:

Reviewer #1:

I reviewed the manuscript submitted by Valenti et al titled - The neurosurgical immigrant experience in Italy: analysis of a northeast tertiary center. This manuscript represents a neurosurgical care analysis of the access of immigrants to a tertiary center in northeastern Italy and the differences in the utilization of health care by immigrant compared to native Italian populations. This paper shows significant demographic & diagnostic disparities present. This is pointing out that the immigrant patients are younger, with a higher incidence of spinal pathologies & emergencies. This is a significant observation to be reported. It also highlights that immigrants, especially those coming from Asia and Central/Eastern Europe; have higher follow-up visit rates, which may reflect potential communication problems or unfamiliarity with the health system. The study underlines how complex the needs of immigrant healthcare are, pointing at cultural, linguistic, and socioeconomic factors influencing access and results. It is well-pegged with data and points to some critical challenges on gaps in integration for recent immigrants, including higher rates of utilization of emergency care as a substitute for primary care.
I must say that this study is quite broad, but it has its limitation in not being able to include irregular immigrants or stratify data based on duration of stay, which could further contextualize the trends of integration.
The manuscript would be further enhanced with minor clarifications on statistical methodologies and further exploration of cultural barriers.
Authors are advised to take care of these comments to make the article better.

  • We appreciate the reviewers for their valuable suggestions. We acknowledge the limitations you highlighted and have tried to improve our work accordingly. Based on your observations, we have added new details as follows:
  • We addressed the lack of data on irregular immigrants in pages 14-15, lines 492 -500;
  • We clarified the statistical methodologies in the statistical analysis section, page 7, lines 200-212;
  • We elaborated on the discussion about cultural barrier on: page 2, lines 50-56; page 6-7, lines 186-197; and page 12-13, lines 406-417;

Reviewer 2 Report

Comments and Suggestions for Authors

1.     The abstract effectively summarizes the study's objectives, methods, and key findings. Sentence structure is clear, but some phrases are repetitive or could be condensed for brevity. 

a.     Include more specific data points, such as exact percentages or numerical results, to strengthen the abstract.

b.     Expand on the conclusion to emphasize actionable recommendations for addressing disparities.

2.     The introduction highlights the growing immigrant population in Italy and the critical need for healthcare access. It provides relevant statistics (e.g., increase from 1.3 million in 2002 to 5.2 million in 2019). 

a.     Expand the discussion of systemic barriers, such as linguistic and cultural differences, and how these may affect neurosurgical care.

b.     Mention why neurosurgical care is a critical focus, considering its urgency and complexity compared to other medical services.

c.     Add global context by discussing immigrant healthcare challenges in other countries, drawing parallels to Italy.

3.     Data collection is described clearly, with details on patient demographics, diagnosis classification, and stratification by geographical origin. Inclusion of ICD-9 codes and the differentiation between inpatient and outpatient data enhances transparency. Some phrases are verbose, e.g., "Patients were grouped according to their geographical origin" could be simplified to "Patients were categorized by geographical origin."

a.     Discuss how potential biases (e.g., underrepresentation of irregular immigrants) were mitigated during data collection.

b.     Provide additional details on the rationale for grouping immigrants by geographic origin and the relevance of these groups to healthcare disparities.

c.     Clarify how the distinction between inpatient and outpatient care reflects differences in healthcare access and utilization.

4.     Results are well-structured, with demographic data and clinical characteristics presented clearly. Significant findings include: Higher rates of spinal pathologies among immigrant populations, Younger age distribution in immigrant inpatients compared to Italians, and Immigrants requiring more frequent follow-up visits. Tables effectively summarize key data but could include confidence intervals for more robust interpretation. Sentences could be streamlined. Example: "Italian patients made up 89.9% of inpatient visits, while immigrants accounted for 10.1%" can be rewritten as "89.9% of inpatients were Italian, with 10.1% being immigrants."

a.     Provide more in-depth analysis of why immigrants from specific regions (e.g., Asia, South America) might face greater challenges in accessing outpatient care.

b.     Discuss how cultural perceptions of illness and healthcare could influence the observed patterns.

c.     Include more granular data on healthcare outcomes (e.g., surgery success rates, patient satisfaction) to assess quality of care.

5.     The discussion provides a balanced interpretation of the results, situating them within the broader literature on immigrant healthcare disparities. Highlights include: Sociocultural barriers limiting immigrant access to outpatient services, The impact of integration time on healthcare utilization, and Higher emergency care usage by immigrants due to unmet primary care needs. Sentence complexity could be reduced. Example: "This may translate to a higher percentage of patients accessing emergency care directly instead of relying on their general practitioner or a lower percentage of these patients accessing regular outpatient visits" could be simplified to "This likely results in higher emergency care use and lower outpatient service utilization."

a.     Expand on the potential impact of employment type (e.g., physical labor among immigrants) on the prevalence of spinal conditions and recurrent visits.

b.     Provide more concrete recommendations for addressing disparities, such as: implementing cultural competence training for healthcare providers, establishing immigrant-focused outpatient clinics, introducing multilingual patient navigators or telemedicine services…etc

c.      Discuss the implications of emergency care overutilization, including cost burdens on the healthcare system and suboptimal patient outcomes.

d.     Examine the potential long-term consequences of unmet healthcare needs among immigrant populations (e.g., chronic disease progression).

e.     Discuss how cultural beliefs, stigma, and trust in the healthcare system might influence immigrant healthcare utilization patterns.

6.     The conclusions summarize key findings, emphasizing the need for targeted interventions to improve immigrant healthcare access. 

a.     Highlight the need for targeted policy interventions, such as improving access to primary and specialist outpatient care for immigrant populations.

b.     Propose strategies to address the unmet needs of immigrants, such as community outreach programs or partnerships with non-governmental organizations.

c.     Suggest strategies for improving immigrant access to outpatient care and reducing reliance on emergency services.

Author Response

Thanks for reviewing our manuscript: The neurosurgical immigrant experience in Italy: analysis of a northeast tertiary center.. Manuscript number: Healthcare-3438873

We have made each of the changes suggested by the reviewers in red and bold, and a copy of the revised manuscript is enclosed for your review.

Please see below for a point-by-point response to the reviewer’s comments:

Reviewer #2:

We deeply appreciate your revisions and the suggestions you provided. We acknowledge the limits you highlighted and have tried to enhance our work. Given your observations, we have incorporated additional details into our text. Here a point to point response to reviewer's comment:

  1. The abstract effectively summarizes the study's objectives, methods, and key findings. Sentence structure is clear, but some phrases are repetitive or could be condensed for brevity.
  2. Include more specific data points, such as exact percentages or numerical results, to strengthen the abstract.

            More patients presented with acute spine pain (26.4% versus 19.6%, p-value: 0.001) as inpatients, while outpatients presented more for degenerative spine issues (77.1% versus 69.0%, p-value: <0.001) but less for brain neoplasms (p-value: 0.009). Additional visit rates were higher for immigrants than Italians (IRR 1.32 visit- year, 95% CI 0.99–1.77 visits - year, p-value: 0.06), especially when spinal (spinal versus cranial: 1.27 visits - year, 95% CI 1.14-1.43 visit-year, p-value: <0.001) and younger (<65 years old: 1.52 visits - year, 95% CI 1.39–1.71 visits per year, p-value: <0.001) patients. [page 1, lines 23, 24, 26-28]

  1. Expand on the conclusion to emphasize actionable recommendations for addressing disparities.

            A healthcare policy tailored to the needs of immigrants—taking into account their cultural and social backgrounds and ensuring effective communication—can be highly beneficial. Specifically, it is essential to reintegrate general practitioners and guiding individuals toward the most appropriate services. [page 1, lines 32-35]

  1. The introduction highlights the growing immigrant population in Italy and the critical need for healthcare access. It provides relevant statistics (e.g., increase from 1.3 million in 2002 to 5.2 million in 2019).
  2. Expand the discussion of systemic barriers, such as linguistic and cultural differences, and how these may affect neurosurgical care.

            Language and the structure of healthcare systems can significantly impact the ability to meet the needs of immigrants. These needs are complex and can vary depending on who is defining them and the potential for recall bias. Self-reported levels of unmet healthcare needs are often influenced by the most significant or distressing memories individuals have. This situation is commonly observed in other countries due to cultural, socio-economic, and communication barriers that arise from different habits and practices [5,6,7]. [page 2, lines 50-56]

  1. Mention why neurosurgical care is a critical focus, considering its urgency and complexity compared to other medical services.

            Discussing healthcare comparing immigrants and native people in neurosurgery appears quite interesting because it is a very specialized sector. Furthermore, every centre can provide a wide range of patients with the same criteria in diagnosis, treatment, conservative or surgical, and follow-up. At the same time, Neurosurgery deals with the brain and the spine, so with a kind of disease that is perceived as urgent and unpreventable. Immigrants are not accustomed to discussing health problems with their general practitioner. So in failing with adequate prevention, neurosurgical evaluations multiply unnecessarily, either as an outpatient or inpatient, through the Emergency Department. [pages 2-3, lines 70-58]

  1. Add global context by discussing immigrant healthcare challenges in other countries, drawing parallels to Italy.

            This situation is commonly observed in other countries due to cultural, socio-economic, and communication barriers that arise from different habits and practices [5,6,7]. [page 2, lines 54-56]

               These macro groups, between the included countries, share similarities in cultural, socio-economic, and linguistic backgrounds, enabling synthetic yet effective data collection. [page 6, lines 179-181]

  1. Data collection is described clearly, with details on patient demographics, diagnosis classification, and stratification by geographical origin. Inclusion of ICD-9 codes and the differentiation between inpatient and outpatient data enhances transparency. Some phrases are verbose, e.g., "Patients were grouped according to their geographical origin" could be simplified to "Patients were categorized by geographical origin."
  2. Discuss how potential biases (e.g., underrepresentation of irregular immigrants) were mitigated during data collection.

            A significant limitation of this study is that we could not conduct a more detailed sub-analysis of sub-pathology based on ICD-9 codes, which could have explored interactions with factors such as country of origin, age, and sex. Additionally, the data abstraction did not include socioeconomic status and patients' income. The inability to perform further analysis considering these modifiers and confounders may have hindered a deeper understanding of the overall analysis. Additionally, a significant limitation is that we could not include irregular immigrants in our analysis. In Italy, once immigrants enter the country to live, they acquire an identification code that uniquely identifies them within the National Healthcare System and for other services, including taxation. We utilized this identification code to trace the original nationality of immigrants, which means that these are all regularly registered immigrants. Once they are registered with a national identification number, it is irrelevant if they migrate as irregular or regular, and they have access to the same healthcare.  Therefore, no irregular immigrants were included, or better, only regularized ones were fully extracted.

Also, we could not track the duration of stay for these immigrants in our country. This is because our healthcare system does not keep records of this information during inpatients or outpatient visits. Our healthcare system is based on the principles of the Italian Constitution and is funded by citizens’ taxes. Therefore, there is no requirement for insurance, and we have no interest in knowing a patient's disposable income, except for those living abroad, who are not the focus of this study. [pages 14-15, lines 487-506]

  1. Provide additional details on the rationale for grouping immigrants by geographic origin and the relevance of these groups to healthcare disparities.

            These macro groups, between the included countries, share similarities in cultural, socio-economic, and linguistic backgrounds, enabling synthetic yet effective data collection. [page 6, lines 179-181]

  1. Clarify how the distinction between inpatient and outpatient care reflects differences in healthcare access and utilization.

            Inpatients typically suffer from severe medical conditions that progress quickly, necessitating urgent hospitalization and prompt diagnosis, which may include surgical intervention. In contrast, outpatients usually have conditions that progress more slowly, allowing for treatment to be scheduled later based on their position on the waiting list and the nature of their disorders. [page 6, lines 164-168]

  1. Results are well-structured, with demographic data and clinical characteristics presented clearly. Significant findings include: Higher rates of spinal pathologies among immigrant populations, Younger age distribution in immigrant inpatients compared to Italians, and Immigrants requiring more frequent follow-up visits. Tables effectively summarize key data but could include confidence intervals for more robust interpretation. Sentences could be streamlined. Example: "Italian patients made up 89.9% of inpatient visits, while immigrants accounted for 10.1%" can be rewritten as "89.9% of inpatients were Italian, with 10.1% being immigrants."
  2. Provide more in-depth analysis of why immigrants from specific regions (e.g., Asia, South America) might face greater challenges in accessing outpatient care.

            Cultural habits influence the perceived needs in health and subsequent decisions [5]. Immigrant women, particularly those from Asian and South Asian backgrounds, as well as practicing Muslims, seek female physicians for their healthcare needs [8,9,10]. They prefer physicians from the same ethnic background, thinking they better understand cultural and religious norms [10]. Immigrant women from South Asia, Eastern Europe, Africa, and Central and South America often feel shy and uncomfortable during physical exams due to their adherence to traditional values regarding modesty and the exposure of body parts [11,12]. Sometimes immigrant women (from Asia, South Asian, and Muslim) do not feel comfortable sharing their health problems with their family members or friends due to fears of being ostracized. In many countries having a severe disease or mental illness is interpreted as a matter of shame and disgrace to the family [13,14]. [pages 6-7, lines 186-197]

  1. Discuss how cultural perceptions of illness and healthcare could influence the observed patterns.

                        Cultural habits influence the perceived needs in health and subsequent decisions [5]. Immigrant women, particularly those from Asian and South Asian backgrounds, as well as practicing Muslims, seek female physicians for their healthcare needs [8,9,10]. They prefer physicians from the same ethnic background, thinking they better understand cultural and religious norms [10]. Immigrant women from South Asia, Eastern Europe, Africa, and Central and South America often feel shy and uncomfortable during physical exams due to their adherence to traditional values regarding modesty and the exposure of body parts [11,12]. Sometimes immigrant women (from Asia, South Asian, and Muslim) do not feel comfortable sharing their health problems with their family members or friends due to fears of being ostracized. In many countries having a severe disease or mental illness is interpreted as a matter of shame and disgrace to the family [13,14]. [page 6-7, lines 186-197]

  1. Include more granular data on healthcare outcomes (e.g., surgery success rates, patient satisfaction) to assess quality of care.

            Dear reviewers, thank you for the suggestion. Unfortunately, we do not have such data. We will add those in future analysis.

  1. The discussion provides a balanced interpretation of the results, situating them within the broader literature on immigrant healthcare disparities. Highlights include: Sociocultural barriers limiting immigrant access to outpatient services, The impact of integration time on healthcare utilization, and Higher emergency care usage by immigrants due to unmet primary care needs. Sentence complexity could be reduced. Example: "This may translate to a higher percentage of patients accessing emergency care directly instead of relying on their general practitioner or a lower percentage of these patients accessing regular outpatient visits" could be simplified to "This likely results in higher emergency care use and lower outpatient service utilization."
  2. Expand on the potential impact of employment type (e.g., physical labor among immigrants) on the prevalence of spinal conditions and recurrent visits.

            Immigrants often face a higher prevalence of manual labor or low-skilled jobs, temporary or informal employment, and lower wages. They tend to experience greater physical demands, poorer environmental working conditions, and increased exposure to occupational risks, particularly ergonomic and psychosocial hazards. Additionally, they generally have worse overall and mental health, as well as challenging employment and working conditions, a situation that the economic crisis has exacerbated. [17,18]. All of these characteristics are quite common in most of the Western countries [5,6]. [page 11, lines 364-371]

  1. Provide more concrete recommendations for addressing disparities, such as: implementing cultural competence training for healthcare providers, establishing immigrant-focused outpatient clinics, introducing multilingual patient navigators or telemedicine services…etc

            There is often a lack of knowledge about immigrants' health needs, which frequently arises from communication barriers. Access to interpreter services, along with a deeper understanding of health practices in their home countries—particularly in developing nations—can significantly enhance the effectiveness of medical interventions. This improvement is significant during residency programs.

            Health policies must address the diverse needs of immigrants. Collaborating with non-governmental organizations (NGOs) can help achieve this goal, as long as these organizations employ personnel trained to understand immigrants' cultural, socio-economic, and educational backgrounds. Integrating immigrants into society is crucial for guiding them toward appropriate health services.

            Schools can play a vital role by offering mandatory health courses that educate younger generations of immigrants. This knowledge can then be shared within families, benefiting older relatives and those recently arriving in Italy. Additionally, family doctors are essential in directing patients to the appropriate exams and specialist visits, which can help reduce reliance on emergency care [24].[page 14, lines 452-466]

  1. Discuss the implications of emergency care overutilization, including cost burdens on the healthcare system and suboptimal patient outcomes.

            Emergency Departments are designed to handle acute symptoms but often do not assess the overall clinical picture. This results in patients frequently returning with the same symptoms without a thorough outpatient diagnostic process. Such repetition can lead to increased healthcare costs, borne by taxpayers in Italy, and longer waiting times in Emergency Departments [33,35].

            Moreover, this reliance on emergency care often exacerbates health issues among immigrants, leading to silent disease progression, such as tumors, chronic pain in spinal disorders, and a greater need for invasive surgeries in cases of severe fractures. Complex health conditions can result in incomplete recovery or increased surgical complications due to decreased mobility, affecting patients' ability to return to their daily activities. [page 14, lines 467-477]

  1. Examine the potential long-term consequences of unmet healthcare needs among immigrant populations (e.g., chronic disease progression).

            Research indicates that immigrants often have better health than native residents when assessed through self-reports and health indicators upon their arrival in the host country. However, over time, this phenomenon known as the "healthy migrant effect" diminishes, and their health tends to decline. [29,30]. Some immigrants are also susceptible to poor health due to exposure to numerous health hazards before, during, and after immigration [6]. [page 13, lines 419-424]

  1. Discuss how cultural beliefs, stigma, and trust in the healthcare system might influence immigrant healthcare utilization patterns.

            Cultural habits influence the perceived needs in health and subsequent decisions [5]. Immigrant women, particularly those from Asian and South Asian backgrounds, as well as practicing Muslims, seek female physicians for their healthcare needs [8,9,10]. They prefer physicians from the same ethnic background, thinking they better understand cultural and religious norms [10]. Immigrant women from South Asia, Eastern Europe, Africa, and Central and South America often feel shy and uncomfortable during physical exams due to their adherence to traditional values regarding modesty and the exposure of body parts [11,12]. Sometimes immigrant women (from Asia, South Asian, and Muslim) do not feel comfortable sharing their health problems with their family members or friends due to fears of being ostracized. In many countries having a severe disease or mental illness is interpreted as a matter of shame and disgrace to the family [13,14]. [pages 6-7, lines 186-197]

  1. The conclusions summarize key findings, emphasizing the need for targeted interventions to improve immigrant healthcare access.
  2. Highlight the need for targeted policy interventions, such as improving access to primary and specialist outpatient care for immigrant populations.

            This highlights the importance of basic health education provided by schools and general practitioners. An effective health policy should encourage investments to enhance the socioeconomic conditions of immigrants, in collaboration with non-governmental organizations. [page 15, lines 530-533]

  1. Propose strategies to address the unmet needs of immigrants, such as community outreach programs or partnerships with non-governmental organizations.

            This highlights the importance of basic health education provided by schools and general practitioners. An effective health policy should encourage investments to enhance the socioeconomic conditions of immigrants, in collaboration with non-governmental organizations. [page 15, lines 530-533]

  1. Suggest strategies for improving immigrant access to outpatient care and reducing reliance on emergency services.

               This highlights the importance of basic health education provided by schools and general practitioners. An effective health policy should encourage investments to enhance the socioeconomic conditions of immigrants, in collaboration with non-governmental organizations. [page 15, lines 530-533]

Reviewer 3 Report

Comments and Suggestions for Authors

The article examines differences in neurosurgical healthcare access between regular immigrants and native Italians at a tertiary hospital in Treviso, Italy, from 2017 to 2021. Analyzing inpatient and outpatient records, the study finds that immigrants, particularly from Asia and Africa, are more likely to seek emergency care rather than scheduled outpatient visits, possibly due to unconsolidated health habits or barriers in primary care access. Immigrants also have a higher rate of additional visits than Italians, especially for spinal conditions and younger patients, suggesting challenges in continuity of care. The study concludes that Central and Eastern European immigrants appear more integrated into the healthcare system, whereas newer immigrant populations may experience structural or cultural barriers.

1.       The exclusion of irregular immigrants significantly reduces the generalizability of the study. Given that irregular immigrants often face the greatest healthcare barriers, their exclusion omits a key demographic.

2.       While ICD-9 codes were used for classification, there is no clear explanation of how diagnoses were confirmed or whether diagnostic accuracy differed between immigrants and Italian patients.

3.       The study does not account for socioeconomic status, employment conditions, or insurance coverage, which could impact healthcare access more than nationality alone.

4.       While the study uses Poisson regression, it does not adequately control for comorbidities, duration of residency in Italy, or health literacy, which are crucial for understanding healthcare utilization disparities.

5.       The study groups all regular immigrants together without distinguishing between long-term and newly arrived immigrants, which could lead to misleading interpretations.

6.       The authors suggest that Central and Eastern European immigrants are "fully integrated," but this conclusion is speculative and not based on qualitative assessments (e.g., patient satisfaction surveys).

7.       Differences in health-seeking behaviors, trust in the medical system, and reliance on traditional medicine among different immigrant groups are not explored.

Author Response

Thanks for reviewing our manuscript: The neurosurgical immigrant experience in Italy: analysis of a northeast tertiary center.. Manuscript number: Healthcare-3438873

We have made each of the changes suggested by the reviewers in red and bold, and a copy of the revised manuscript is enclosed for your review.

Please see below for a point-by-point response to the reviewer’s comments:

Reviewer #3:

The article examines differences in neurosurgical healthcare access between regular immigrants and native Italians at a tertiary hospital in Treviso, Italy, from 2017 to 2021. Analyzing inpatient and outpatient records, the study finds that immigrants, particularly from Asia and Africa, are more likely to seek emergency care rather than scheduled outpatient visits, possibly due to unconsolidated health habits or barriers in primary care access. Immigrants also have a higher rate of additional visits than Italians, especially for spinal conditions and younger patients, suggesting challenges in continuity of care. The study concludes that Central and Eastern European immigrants appear more integrated into the healthcare system, whereas newer immigrant populations may experience structural or cultural barriers.

            We deeply thank the reviewers for their precious insight and the suggestions provided. We agree with the limitations you highlighted, and we have made improvements. Considering your observations, we have added new details to our work. Here is a point to point response to your valuable comments:

  1. The exclusion of irregular immigrants significantly reduces the generalizability of the study. Given that irregular immigrants often face the greatest healthcare barriers, their exclusion omits a key demographic.

            Additionally, a significant limitation is that we could not include irregular immigrants in our analysis. In Italy, once immigrants enter the country to live, they acquire an identification code that uniquely identifies them within the National Healthcare System and for other services, including taxation. We utilized this identification code to trace the original nationality of immigrants, which means that these are all regularly registered immigrants. Once they are registered with a national identification number, it is irrelevant if they migrate as irregular or regular, and they have access to the same healthcare.  Therefore, no irregular immigrants were included, or better, only regularized ones were fully extracted. [pages 14-15, lines 492-500]

  1. While ICD-9 codes were used for classification, there is no clear explanation of how diagnoses were confirmed or whether diagnostic accuracy differed between immigrants and Italian patients.

            The diagnosis was made using the same tools applied to the Italian patients: first, we investigated the patient’s medical history, which guided the subsequent physical examination. Then, we clarified diagnostic suspicions with instrumental analysis, such as MRI, CT scans, X-rays, neurophysiology tests, blood tests, and other specialist assessments. [page 6, lines 58-62]

  1. The study does not account for socioeconomic status, employment conditions, or insurance coverage, which could impact healthcare access more than nationality alone.

            Immigrants often face a higher prevalence of manual labor or low-skilled jobs, temporary or informal employment, and lower wages. They tend to experience greater physical demands, poorer environmental working conditions, and increased exposure to occupational risks, particularly ergonomic and psychosocial hazards. Additionally, they generally have worse overall and mental health, as well as challenging employment and working conditions, a situation that the economic crisis has exacerbated. [17,18]. All of these characteristics are quite common in most of the Western countries [5,6]. [page 11, lines 64-71]

            The literature identifies various reasons to explain the differences between native and foreign populations. The most significant barriers include cultural factors, communication issues, socio-economic status, the structure of the healthcare system, and immigrants' knowledge about the healthcare system [5]. Among those there is also perceived discrimination, and racism [25], difficulty accessing the system, longer wait times, lack of direct access to specialists, laboratory or other diagnostic reporting [26]. Cultural factors should not be underestimated, such as the inability to express personal beliefs like previous immigrants did or the need to adapt to certain aspects of a Western lifestyle, including changes in diet and work-leisure balance [27]. Hence, the immigrants tend to use healthcare services less frequently than native residents, terminate treatment early and receive lower-quality healthcare [6]. [page 12-13, lines 406-417]

               Research indicates that immigrants often have better health than native residents when assessed through self-reports and health indicators upon their arrival in the host country. However, over time, this phenomenon known as the "healthy migrant effect" diminishes, and their health tends to decline. [29,30]. Some immigrants are also susceptible to poor health due to exposure to numerous health hazards before, during, and after immigration [6]. [page 13, lines 419-424]

            Also, we could not track the duration of stay for these immigrants in our country. This is because our healthcare system does not keep records of this information during inpatients or outpatient visits. Our healthcare system is based on the principles of the Italian Constitution and is funded by citizens’ taxes. Therefore, there is no requirement for insurance, and we have no interest in knowing a patient's disposable income, except for those living abroad, who are not the focus of this study. [page 15, lines 501-506]

  1. While the study uses Poisson regression, it does not adequately control for comorbidities, duration of residency in Italy, or health literacy, which are crucial for understanding healthcare utilization disparities.

We appreciate the reviewer's insights and apologize for our analysis's lack of clarity and depth. Unfortunately, this analysis did not control for socioeconomic status and employment situations. Due to the abstraction of ICD-9 codes, we could not explore those specific factors. We have mentioned this limitation in the limitations section of our paper. Regarding insurance, the Italian healthcare system provides free care for individuals who cannot afford it, effectively eliminating the need for health insurance coverage.  [page 11, lines 508-513]

  1. The study groups all regular immigrants together without distinguishing between long-term and newly arrived immigrants, which could lead to misleading interpretations.

            Additionally, a significant limitation is that we could not include irregular immigrants in our analysis. In Italy, once immigrants enter the country to live, they acquire an identification code that uniquely identifies them within the National Healthcare System and for other services, including taxation. We utilized this identification code to trace the original nationality of immigrants, which means that these are all regularly registered immigrants. Once they are registered with a national identification number, it is irrelevant if they migrate as irregular or regular, and they have access to the same healthcare.  Therefore, no irregular immigrants were included, or better, only regularized ones were fully extracted.

            Also, we could not track the duration of stay for these immigrants in our country. This is because our healthcare system does not keep records of this information during inpatients or outpatient visits. Our healthcare system is based on the principles of the Italian Constitution and is funded by citizens’ taxes. Therefore, there is no requirement for insurance, and we have no interest in knowing a patient's disposable income, except for those living abroad, who are not the focus of this study.[pages 14-15, lines 492-506]

  1. The authors suggest that Central and Eastern European immigrants are "fully integrated," but this conclusion is speculative and not based on qualitative assessments (e.g., patient satisfaction surveys).

            Language and the structure of healthcare systems can significantly impact the ability to meet the needs of immigrants. These needs are complex and can vary depending on who is defining them and the potential for recall bias. Self-reported levels of unmet healthcare needs are often influenced by the most significant or distressing memories individuals have. This situation is commonly observed in other countries due to cultural, socio-economic, and communication barriers that arise from different habits and practices [5,6,7]. [page 2, lines 50-56]

  1. Differences in health-seeking behaviors, trust in the medical system, and reliance on traditional medicine among different immigrant groups are not explored.

            Cultural habits influence the perceived needs in health and subsequent decisions [5]. Immigrant women, particularly those from Asian and South Asian backgrounds, as well as practicing Muslims, seek female physicians for their healthcare needs [8,9,10]. They prefer physicians from the same ethnic background, thinking they better understand cultural and religious norms [10]. Immigrant women from South Asia, Eastern Europe, Africa, and Central and South America often feel shy and uncomfortable during physical exams due to their adherence to traditional values regarding modesty and the exposure of body parts [11,12]. Sometimes immigrant women (from Asia, South Asian, and Muslim) do not feel comfortable sharing their health problems with their family members or friends due to fears of being ostracized. In many countries having a severe disease or mental illness is interpreted as a matter of shame and disgrace to the family [13,14]. [pages 6-7, lines 186-197]

Round 2

Reviewer 2 Report

Comments and Suggestions for Authors

This is a valuable and well-conducted study. Having addressed the previous suggestions, significantly enhanced the manuscript's quality and impact.

Reviewer 3 Report

Comments and Suggestions for Authors

Thank you to the authors. I think it is suitable for publication in its current form.